# Palladium(II) and Platinum(II) Complexes Bearing ONS-Type Pincer Ligands: Synthesis, Characterization and Catalytic Investigations

**DOI:** 10.3390/molecules29143425

**Published:** 2024-07-22

**Authors:** Alfonso Castiñeiras, Isabel García-Santos

**Affiliations:** Department of Inorganic Chemistry, Faculty of Pharmacy, University of Santiago de Compostela, 15782 Santiago de Compostela, Spain; isabel.garcia@usc.es

**Keywords:** thiosemicarbazone complexes, ONS pincer ligands, Pd(II) and Pt(II) complexes, homogeneous catalysis, C–C bond formation

## Abstract

This work describes the synthesis of eight new Pd(II) and Pt(II) complexes with the general formula [M(TSC)Cl], where TSC represents the 4N-monosubstituted thiosemicarbazone derived from 2-acetylpyridine N-oxide with the substituents CH_3_ (H4MLO), C_2_H_5_ (H4ELO), phenyl (H4PLO) and (CH_3_)_2_ (H4DMLO). These complexes have been characterized by elemental analysis, molar conductivity, IR spectroscopy, ^1^H, ^13^C, ^195^Pt and ESI-MS. The complexes exhibit a square planar geometry around the metallic center coordinated by a thiosemicarbazone molecule acting as a donor ONS-type pincer ligand and by a chloride, as confirmed by the molecular structures of the complexes, [Pd(4ELO)Cl] (**3**) and [Pd(4PLO)Cl] (**5**), determined by single-crystal X-ray diffraction. The ^195^Pt NMR spectra of the complexes of formulae [Pt(4PLO)Cl] (**6**) and [Pt(4DMLO)Cl] (**8**) in DMSO show a single signal at −2420.4 ppm, confirming the absence of solvolysis products. Complexes **3** and **5** have been tested as catalysts in the Suzuki–Miyaura cross-coupling reactions of aryl bromides with phenylboronic acid, with yields of between 50 and 90%

## 1. Introduction

For many years, catalysis promoted by transition metal ion complexes has been an important field of research, both in basic science and applied science, because it plays a valuable role in the synthesis of fundamental products in the petrochemical, pharmaceutical, food or cosmetics industries, among others. In addition, in the last twenty years, the concept of sustainable chemistry has increased the interest in maximizing the ecological factors in the design of new synthesis processes as a result of the challenges imposed by current society related to the availability of natural resources and with the environmental and climatic factors. In this sense, the mediated reactions by transition metal complexes acting as catalysts are usually very clean, consuming the maximum proportion of raw materials, and the waste products are minimal, generating substances with little toxicity for living beings and the environment, which is concordant with some of the precepts of green chemistry [1]. Consequently, this evolution has caused, in recent years, that the research on homogeneous catalysis using transition metal complexes in the laboratory and the search for better catalysts are continuous. The reactions that were believed to be well understood and optimized have now been revolutionized with completely new catalysts and a selection of unique products.

An important group of catalysts in metal–organic chemistry, accepted by the pharmaceutical, fine chemistry and energy industries [2], are the complexes with pincer-type ligands, which are used in several applications such as the transformation and synthesis of various organic compounds, including amines, pyridines and the derivatives of carboxylic acids [3,4]. The pincer-type ligands are chelating agents that bind strongly to three adjacent co-planar positions of a metal complex to form two metallocyclic rings of five members [5,6].

A group of compounds that can act as pincer-type ligands are certain thiosemicarbazones. In general, thiosemicarbazones (TSCs) represent a very interesting category of organic compounds at the research level due to their versatile chemical properties and their wide range of applications, thanks to their use as precursors in organic synthesis, as coordination chemistry ligands and the considerable therapeutic and biological activity presented by some of them or their derivatives.

The use of TSCs as ligands in the formation of metal complexes has been known for many years. The simplest TSCs contain the chemically active chain, >C=N–N(H)–(=S)–N<, with four potential donors, represented by the nitrogen atoms azometine (N_az_), hydracine (N_hy_), thioamide (N_ta_) and the sulfur thioamide atom (S), with a coordination capacity attributed to electronic delocalization throughout the chain, resulting in a wide variety of coordination modes. The aforementioned chain is usually almost flat with the atom of S anti-N_az_ in the solid state [7]. However, TSCs adopt a conformation with the sin-N_az_ atom, when coordinated with metallic ions. In a solid phase, TSCs are known in their thione, but in solution they exist in their thiol form, with a thione–thiol tautomer equilibrium. In general, TSCs coordinate as neutral ligands (thione form) or monodesprotonated (thiolate form) through S and N_hy_, forming a highly stable five-member ring. Additionally, the denticity degrees are increased when functional groups containing donor atoms, such as pyridines, carbonyls, etc., are attached to the azomethine carbon atom in the vicinity of the donor atoms of the TCS. These functionalized tiosemicarbazonas, incorporating a new donor atom, can act as pincer-type ligands, giving rise, in the presence of certain transition metals ions, to structurally predictable and highly robust complexes, given that complexation usually features two metallocyclic rings of five members, or one of five and one of six members. This behavior is important when it comes to carbon–carbon coupling reactions and especially those that involve palladium complexes, since it is known that pincer-type Pd(II) complexes efficiently catalyze many of these reactions, having raised the hypothesis that the tridentada coordination of the pincer-type ligands stabilizes the metal–carbon bond during the catalytic cycle [8].

Suzuki–Miyaura coupling or Suzuki coupling is the name given to metal-catalyzed reactions, often palladium, which occur between an organoborane with an alkyl, aryl or vinyl group and a halogenated derivative (aryl or vinyl halides) under basic conditions, and it is one of the palladium-catalyzed C–C, C–N and C–O bond-forming processes. The key aspects of the cross-coupling reaction of Suzuki–Miyaura are, among others, the tolerance towards functional groups, the availability and low toxicity of the starting reagents, the stability of boronic acids toward heat, oxygen and water, and the ease of manipulation and separation of the boron present in the by-products contained in the reaction mixtures [9,10,11].

Although transition metal complexes with thiosemicarbazones as ligands have been known for a long time, the use of some of them in homogeneous catalysis and their study are quite recent. In particular, the reports of catalytic cross-coupling reactions with such complexes have appeared only in the last 20 years [12]. In this sense, in recent years, we have studied the behavior as catalysts of various palladium(II) complexes with thiosemicarbazones against the Suzuki–Miyaura cross-coupling reaction [13,14]. We now report on the synthesis of eight new Pd(II) and Pt(II) complexes with the tiosemicarbazonas 4N-substituted derivatives of 2-acetylpyridine N-oxide. The complexes have been characterized via elementary analysis, FTIR, UV-VIS, Multinuclear NMR, molar conductivity and ESI-MS. The tiosemicarbazonas act as pincer-type ligands in an ONS coordination mode, as confirmed by the structures of [Pd(4ELO)Cl] (**3**) and [Pd(4PLO)Cl] (**5**), as determined by the X-ray diffraction of single crystals. In addition, the catalytic activity of complexes **3** and **5** was explored in the Suzuki–Miyaura cross-coupling reaction of aryl-bromide with phenylboronic acid with good yields.

## 2. Results

### 2.1. Synthesis and Characterization

In the present work, four 2-acetylpyridine-*n*-thiosemicarbazones 4N-substituted derivatives have been used, differing in the substituent on the nitrogen atom thioamide (R = Me, Et, Ph and 2Me). The reaction of these thiosemicarbazones with K_2_[PdCl_4_] or K_2_[PtCl_4_] in ethanol/water 2:1, maintained at reflux, gave two groups of complexes of the general formulae, [Pd(TSC)Cl] (**1**, **3**, **5**, **7**) and [Pt(TSC)Cl] (**2**, **4**, **6**, **8**), with an average yield of 68% (Figure 1). The compounds were well characterized via various spectroscopic techniques, and the molecular structures of compounds **3** and **5** were determined via single-crystal X-ray diffraction. All the compounds have a square planar geometry with monodeprotonated ligands coordinating in the ONS pincer-type tridentate mode. The complexes exhibit moderate solubility in common organic solvents, such as methanol, ethanol, acetonitrile, dimethyl sulfoxide (DMSO) and dimethyl formamide (DMF). However, they are poorly soluble or insoluble in carbon tetrachloride, chloroform and dichloromethane.

The molar conductance values observed for all of the complexes are between 6 and 20 mS cm^2^ mol^−1^ at 25 °C in DMF at a concentration of 10^−3^ mol L^−1^. This is below the expected value for a 1:1 electrolyte (60–80 mS cm^2^ mol^−1^) [15]. In the mass spectra, the appearance of the peak corresponding to the molecular ions of complexes **3**, **4**, **7** and **8** supports their 1:1 (M:L) stoichiometry. Furthermore, the spectra of all complexes show a fragment of the [MH^+^-Cl] ion suggesting that the M-L bond is weak. The mass spectra (Appendix A), IR spectra (Appendix A), ultraviolet–visible spectra (Appendix A), ^1^H NMR spectra (Appendix A), ^13^C NMR spectra (Appendix A) and ^195^Pt NMR spectra (Appendix A) are provided in the Appendix A for further examination.

### 2.2. FT-IR Spectra

The infrared spectra of the isolated TSCs reveal the presence of two medium-intensity bands at 3315 cm^−1^ and 3030 cm^−1^, which have been attributed to the ν(NH) vibrations. The first band, which corresponds to the N_ta_–H bond, remains in a position close to 3325 cm^−1^ in the complexes, while the second, due to N_hy_–H, disappears as a result of the enolination prior to deprotonation and coordination through the thiolate sulfur atom. This is confirmed by the shift to lower frequencies of the assigned band ν(C=S), which appears in ligands at 840–800 cm^−1^. This displacement is due to ν(C–S) and, consequently, to coordination by the S atom [16,17,18,19]. Similarly, a very strong band between 1520 and 1550 cm^−1^, which was previously assigned to the C=N stretching modes in the complexes is shifted to lower frequencies (1495–1485 cm^−1^). This shift confirms the coordination of metals through the azomethine nitrogen atom. Another absorption band, between 1256 and 1223 cm^−1^, is characteristic of the N–O stretching modes of free TSCs. In all complexes, this band shifts to lower frequencies (1185–1190 cm^−1^), confirming the coordination of the oxygen atom from pyridine N-oxide to the metal. This is also reinforced by the assignment of ν(MO), ν(MN), ν(MS) and ν(MCl) in the far IR, which appear approximately around 409, 445, 334 and 317 cm^−1^ in the Pd complexes and at 429, 443, 331 and 317 cm^−1^ in the Pt complexes, respectively. These values are consistent with those reported in the literature [20].

### 2.3. UV–Visible Spectra

All the complexes are diamagnetic according to the assigned plane-square geometry. The electronic spectra of the palladium(II) complexes show three absorption bands in the 26,500–20,500 cm^−1^ region. The bands appearing in the region of 26,500–25,000 cm^−1^ were assigned to intraligand transitions, while those appearing around 24,300–23,275 cm^−1^ were assigned to LMCT, and the most intense between 20,900 and 22,800 cm^−1^ are a combination of S→Pd(II) and ^1^A_1g_→^1^B_1g_ transitions [21]. The electron spectra of the platinum(II) complexes show a band between 27,000 and 25,900 cm^−1^ assigned to the ^1^A_1g_→^1^A_2g_ transition. A second band at 24,700–23,400 cm^−1^ was assigned to the ^1^A_1g_→^1^E_g_ transition, and two weak bands at 18,000 and 16,000 cm^−1^ were assigned to forbidden spin transitions [22].

### 2.4. NMR Spectra of the Complexes

The NMR signals of the complexes in DMSO were identified according to the published data [16,17,18,19].

The ^1^H NMR spectra of the complexes, in DMSO, present a singlet at δ 2.47–2.65 ppm that corresponds to the methyl protons of the –C(CH_3_)=N– group, shifted to a high field in relation to the same signal in the free ligands (δ 2.23–2.37 ppm), due to the electronic delocalization and increase in the bond order as a consequence of the coordination to the metal through the azamethine nitrogen atom. The singlet that in the free ligands appears around 10.50 ppm due to the proton on the hydracinic nitrogen atom. In the complexes, this signal has disappeared, as a consequence of the coordination of the TSC to the metal, in its monoanionic form, through the atom of thiolate sulfur, after thione→thiol tautomerism during the complexation and deprotonation of said hydrazine atom. In the complexes where the TSC is 4N-monosubstituted, the signal due to the N–H of the thioamide group appears shifted to a high field as a result of a slight increase in the bond order in –C(–S)–NH– caused by the coordination. The downward shifts of most pyridine ring signals may reflect the coordination through the oxygen atom.

The ^13^C NMR spectra of the complexes are similar to those of the TSC ligands although somewhat slightly sensitive to coordination. The methyl carbon atom of the group –C(CH_3_)=N– resonated at δ 13.9–20.1 ppm. The azomethine carbon atom, C=N, in the complexes resonated at 138.5–147.5 ppm, in contrast to the free TSCs that exhibit resonances at δ 139.5–147.5 ppm. The carbon atom of the C–S bond appears at 177.9–184.5 ppm.

The only platinum complexes soluble enough to obtain ^195^Pt NMR spectra were those with the H4PLO and H4DMLO ligands. These spectra show a single signal at −2420.4 ppm, confirming the absence of any type of solvolysis. However, the slight amplitude of such signals may indicate a small dynamic exchange equilibrium between the ligand and the solvent, as has been suggested in similar systems.

### 2.5. Molecular Structures and Supramolecular Analysis

Table 1 summarizes the relevant crystal data and the refinement of the structures of compounds **3** and **5**. Table 2 shows the coordination bond lengths and angles, and Appendix A show the relevant parameters of the hydrogen bonding and ring–ring stacking interactions of the two compounds. The molecular structures of **3** and **5** are similar and will be discussed together. A view of the asymmetrical unit of **3** and **5** is shown in Figure 1a and Figure 1b, respectively. Each structure consists of a single palladium center bound by a tridentate monoanionic pincer ligand 2-acetylpyridine-*n*-oxide 4N-ethylthiosemicarbazonate (4ELO^−^) or 2-Acetylpyridine-*n*-oxide 4N-phenylthiosemicarbazonate (4PLO^−^) along with a chlorido ligand to complete a distorted square planar geometry. The bite angle of the 6- and 5-membered chelate rings of 4.73° and 5.84° are similar to that observed in other palladium(II) complexes with thiosemicarbazone ONS pincer ligands and the bond distances with respect to the palladium center are consistent with the sum of the covalent radii for Pd–Cl, Pd–N, Pd–O and Pd–S (2.27, 2.03, 2.01 and 2.30 Å, respectively) falling in a similar order: Pd(1)–N(12)_aza_ < Pd(1)–O(11)_N-oxide_ < Pd(1)–S(1)_thiolate_. The Pd–Cl, Pd–N, Pd–O and Pd–S distances are roughly equal to the average found in the CCDC Database [23], and are consistent with those found in other complexes with TSC-ONS donators with the same structure such as 2-hydroxybenzaldehyde 4N-ethylthiosemicarbazone at 2.3078(8), 1.965(2), 2.019(2) and 2.2456(9) Å, respectively [24]. As expected, this Pd-S distance is longer than the Pd-Cl distance.

The crystal packing in both complexes is determined by the existence of a hydrogen bond between the N–H thioamide bond as a donor and the chlorido of a nearest neighboring molecule as an acceptor (2.80 Å, 146° and 2.75 Å, 171°, for Cl···H, ∠N–H···Cl in **3** and **5**, respectively), giving rise to chains in the direction of the “c” axis (**3**) and the angle bisector between the axes “a” and “c” (**5**) (Figure 2a,b) (Appendix A). These chains, in turn, are connected through ring–chelate interactions [d_(Cg-Cg)_ = 3.608(2) Å] between the pyridine ring and the five-membered chelate in **3**, through ring–chelate and ring–ring interactions [d(_Cg-Cg)_ = 3.701(2) and 3.623(2) Å, respectively], between the pyridine ring and the five-membered chelate and between the pyridine rings of neighboring molecules in **5** (Figure 2c,d) (Appendix A).

### 2.6. Catalysis

Knowing that many palladium complexes serve as efficient catalysts for C-C cross-coupling reactions [11,25,26], we decided to investigate these catalytic properties in two Pd(II) complexes as representative cases. To this end, the catalytic activity of compounds **3** and **5**, whose molecular structure has been determined via X-ray diffraction, has been analyzed against the Suzuki–Miyaura C-C cross-coupling reaction between phenylboronic acid and the aryl bromides, namely 1-bromo-4-methoxybenzene, 1-bromo-4-methylbenzene and 1-bromo-4-nitrobenzene, to produce the corresponding p-diphenyl derivative (Figure 2). To carry out these reactions, the same reaction conditions were considered as those used in the previous studies with TSCs [14].

Aryl bromides have been selected as representative cases due to the fact that among the three substituents used (R = -OCH_3_, -CH_3_, -NO_2_), methoxy has the greatest electron-donating power and nitro has the greatest electron-withdrawing power. The selected aryl bromides and p-phenylboronic acid were reacted at 120 °C for 24 h using Na_2_CO_3_ (2 mmol) as a base in the presence of a small amount of DMF (3 mL) and using 1 mol% of [Pd(4ELO)Cl] (**3**) or [Pd(4PLO)Cl] (**5**) as catalysts. Each experiment was repeated three times, and the resulting mean values are presented in Table 3. Although the two complexes show comparable catalytic efficiency, the best results were obtained when the aryl bromide substituent was the nitro group. In view of the results obtained, it can be concluded that the catalytic activity depends on the substituent groups in the ring, the electron-withdrawing groups increase the reaction rate and, consequently, in the processes studied, the catalytic activity follows the sequence NO_2_ > CH_3_ > OCH_3_; it is in the derived nitro where the yield was greater than 90%, which has also been observed in similar processes [8,27]. Based on our results, mechanistic investigations and literature reports, the general catalitic cycle for Suzuki coupling is widely accepted to proceed via three basic steps: oxidative addition, transmetallation and reductive elimination.

## 3. Materials and Methods

### 3.1. Physical Measurements

The elemental analyses were carried out on a Fisons-Carlo Erba 1108 microanalyser (CARLO ERBA Reagents SAS, Chaussée du Vexin, Val-de-Reuil, France). The melting points were determined in open tubes using a Büchi apparatus (BUCHI Ibérica, Barcelona, Spain) and are uncorrected. The mass spectra were obtained in a HP5988A spectrometer for EI and a Micromass AUTOSPEC spectrometer (nitrobenzyl alcohol matrix) for FAB (Agilent Technologies, Inc., Santa Clara, CA, USA). IR spectra were recorded as KBr disks (4000–400 cm^−1^) or polyethylene-sandwiched Nujol mulls (500–100 cm^−1^) using a Bruker IFS-66v spectrophotometer (Bruker Corporation, Billerica, MA, USA). The electronic spectra were carried out on a SHIMADZU UV-3101PC spectrophotometer (Izasa Scientific, Barcelona, Spain) equipped with a reflectance accessory. The conductivity measurements were carried out using a CRISON digital conductivity bridge model MicroCM 2202 (Crison Instruments, Alella, Barcelona, Spain) using freshly prepared 10^–3^ M solutions of the complexes in DMF. ^1^H and ^13^C NMR spectra were obtained as DMSO-d_6_ solutions using a Varian Mercury 300 instrument (Varian Medical Systems, Inc, Palo Alto, CA, USA). ^195^Pt NMR spectra in DMSO-d_6_ were run on Bruker AMX 500 apparatus (Bruker Corporation, Billerica, MA, USA). Chemical shifts are expressed on the δ scale (downfield shifts positive) relative to TMS (^1^H and ^13^C spectra) and a 1M aqueous solution of Na_2_PtCl_6_ (^195^Pt spectra).

### 3.2. Synthesis of Thiosemicarbazone Ligands

All the reagents and solvents were purchased from Sigma-Aldrich (Sigma-Aldrich. Inc., Tres Cantos, Madrid, Spain), were of reagent grade and unless otherwise specified, were used as received. The solvents were purified using the conventional methods. The following thiosemicarbazones were prepared as described in the literature: 2-acetylpyridine-*n*-oxide ^4^N-methylthiosemicarbazone (H4MLO) [16], 2-acetylpyridine-*n*-oxide- ^4^N-ethylthiosemicarbazone (H4ELO) [17], 2-acetylpyridine-*n*-oxide ^4^N-phenylthiosemicarbazone (H4PLO) [18] and 2-acetylpyridine-*n*-oxide ^4^N-dimethylthiosemicarbazone (H4DMLO) [19].

### 3.3. Synthesis and Crystallization of the Complexes

In the general procedure for the preparation of the Pd(II) and Pt(II) complexes, a solution of K_2_[PdCl_4_] or K_2_[PtCl_4_] (0.74 mmol) in 10 mL of H_2_O was added to the thiosemicarbazone (0.74 mmol) in ethanol (20 mL). The mixture was refluxed for 1 h and then was stirred at room temperature for 7 days. The resulting solids were filtered off, washed thoroughly with cold ethanol and stored in a desiccator over CaCl_2_, until required for characterization.

*[Pd(4MLO)Cl]* (**1**): Yellow (0.28 g, 56%). FAB^+^ MS [*m*/*z*, assignment]: 329 [Pd(4MLO)]^+^, 223 [4MLO]^+^. C_9_H_11_ClN_4_OPdS (365.12): calcd. C 29.6, H 3.04, N 15.3, S 8.7; found C 29.8, H 3.1, N 15.5, S 8.6. Selected IR data (ν_max_/cm^−1^): 3308 s ν(NH), 1568 w, 1547 m, 1520 m, 1493 s, 1479 s, 1406 s [ν(C=N) + ν(C=C)], 1367 w, 1319 w [ν(C=S) + ν(C=N)], 1282 m, 1267 w, 1230 w, 1190 m ν(NO), 1176 m, 1136 m, 1070 w, 1045 m ν(NN), 833 w ν(C=S), 775m. Far-IR (Nujol, ν/cm^−1^): 456 w ν(Pd–N), 411 w ν(Pd-O), 329 s ν (Pd–S), 314 sh ν(Pd–Cl). UV-Vis, (λ_max_, cm^–1^): 26,420, 24,271, 22,805. Conductance (Λ_m_/μS cm^−1^) in DMF: 17. ^1^H NMR (DMSO, ppm): 8.81 (1H, N4H), 8.16–7.69 (4H, pyH), 3.34 (3H, NCH_3_), 2.61 (3H, CCH_3_). ^13^C (DMSO, ppm): 178.3 (C=S), 144.8 (C=N), 126.7–123.2 (py), 32.0 (NCH_3_), 13.9 (CCH_3_).

*[Pt(4MLO)Cl]* (**2**): Orange (0.45 g, 72%). FAB^+^ MS [*m*/*z*, assignment]: 418 [Pt(4MLO)]^+^, 223 [4MLO]^+^. C_9_H_11_ClN_4_OPtS (453.81): calcd. C 23.8, H 2.44, N 12.3, S 7.07; found C 24.1, H 2.7, N 12.8, S 7.1. Selected IR data (ν_max_/cm^−1^): 3329 s ν(NH), 1563 w, 1539 m, 1516 m, 1493 m, 1479 s, 1406 s [ν(C=N) + ν(C=C)], 1369 m, 1320 w [ν(C=S) + ν(C=N)], 1282 m, 1269 m, 1230 w, 1186 m ν(NO), 1176 m, 1138 m, 1099 w, 1070 w, 1045 m, ν(NN), 837 w ν(C=S), 773 m. Far-IR (Nujol, ν/cm^−1^): 457 w ν(Pt–N), 419 w ν(Pt-O), 333 s ν(Pt–S), 316 s ν(Pt–Cl). UV-Vis, (λ_max_, cm^–1^): 26,075, 23,474, 20,725, 18,066, 17,196, 16,207. Conductance (Λ_m_/μS cm^−1^) in DMF: 6. ^1^H NMR (DMSO, ppm): 8.94 (1H, N4H), 8.27–7.71 (4H, pyH), 3.50 (3H, NCH_3_), 2.56 (3H, CCH_3_). ^13^C (DMSO, ppm): 177.0 (C=S), 143.5 (C=N), 126.9–124.1 (py), 31.6 (NCH_3_), 14.2 (CCH_3_).

*[Pd(4ELO)Cl]* (**3**): Orange (0.26 g, 53%). FAB^+^ MS [*m*/*z*, assignment]: 377 [Pd(4ELO)Cl]^+^, 343 [Pd(4ELO)]^+^, 237 [4ELO]^+^. C_10_H_13_ClN_4_OPdS (379.17): calcd. C 31.7, H 3.46, N 14.8, S 8.5; found C 31.5, H 3.4, N 14.6, S 8.4. Selected IR data (ν_max_/cm^−1^): 3326 m ν(NH), 1563 w, 1545 m, 1498 m, 1493 s, 1411 m [ν(C=N) + ν(C=C)], 1390 s, 1371 m, 1335 m [ν(C=S) + ν(C=N)], 1295 m, 1282 m, 1232 w, 1188 m ν(NO), 1164 w, 1136 m, 1090 w, 1070 w, 1042 m ν(NN), 767 m. Far-IR (Nujol, ν/cm^−1^): 445 w ν(Pd–N), 417 w ν(Pd-O), 339 s ν(Pd–S), 319 w ν(Pd–Cl). UV-Vis, (λ_max_, cm^–1^): 26,420, 23,282, 21,598. Conductance (Λ_m_/μS cm^−1^) in DMF: 14. ^1^H NMR (DMSO, ppm): 8.79 (1H, N4H), 8.16–7.66 (4H, pyH), 3.24 (2H, CH_2_), 2.60 (3H, CCH_3_), 1.10 (3H, CH_3_). ^13^C (DMSO, ppm): 178.2 (C=S), 147.5 (C=N), 143.6–125.6 (py), 39.0 (CH_2_), 18.5 (CCH_3_), 14.5 (CH_3_).

*[Pt(4ELO)Cl]* (**4**): Orange (0.46 g, 78%). FAB^+^ MS [*m*/*z*, assignment]: 468 [Pt(4ELO)Cl]^+^, 432 [Pt(4ELO)]^+^. C_10_H_13_ClN_4_OPtS (467.83): calcd. C 25.7, H 2.80, N 12.0, S 6.9; found C 25.9, H 2.8, N 11.8, S 7.0. Selected IR data (ν_max_/cm^−1^): 3344 m ν(NH), 1564 w, 1539 m, 1494 s, 1481 s, 1410 m [ν(C=N) + ν(C=C)], 1390 s, 1371 m, 1334 m [ν(C=S) + ν(C=N)], 1302 m, 1284 m, 1234 w, 1186m ν(NO), 1138 m, 1089 m, 1070 m, 1041 s ν(NN), 825 w ν(C=S), 765 m. Far-IR (Nujol, ν/cm^−1^): 431 w ν(Pt-N), 332 sh ν(Pt–S), 315 s ν(Pt–Cl). UV-Vis, (λ_max_, cm^–1^): 26,809, 24,691, 21,344, 20,202, 20,992, 15,847, 15,540. Conductance (Λ_m_/μS cm^−1^) in DMF: 15. ^1^H NMR (DMSO, ppm): 8.94 (1H, N4H), 8.27–7.72 (4H, pyH), 3.31 (2H, CH_2_), 2.55 (3H, CCH_3_), 1.12 (3H, CH_3_). ^13^C (DMSO, ppm): 177.9 (C=S), 147.3 (C=N), 142.6–125.9 (py), 39.0 (CH_2_), 19.1 (CCH_3_), 14.6 (CH_3_).

*[Pd(4PLO)Cl]* (**5**): Brown (0.29 g, 63%). FAB^+^ MS [*m*/*z*, assignment]: 391 [Pd(4PLO)]^+^, 286 [4PLO]^+^. C_14_H_13_ClN_4_OPdS (427.21): calcd. C 39.4, H 3.07, N 13.1, S 7.5; found C 39.8, H 3.1, N 13.0, S 7.7. Selected IR data (ν_max_/cm^−1^): 3306 m ν(NH), 1599 w, 1533 m, 1471 s, 1440 s, 1411m [ν(C=N) + ν(C=C)], 1369 w, 1325 m [ν(C=S) + ν(C=N)], 1280 w, 1253 w, 1188 m ν(NO), 1167 w, 1140 m, 1099 w, 1072 w 1043 m ν(NN), 852 w ν(C=S), 750 m. Far-IR (Nujol, ν/cm^−1^): 433 m ν(Pd–N), 400 w ν(Pd-O), 330 s ν(Pd–S), 316 sh ν(Pd–Cl). UV-Vis, (λ_max_, cm^–1^): 25,477, 23,866, 20,942. Conductance (Λ_m_/μS cm^−1^) in DMF: 12. ^1^H NMR (DMSO, ppm): 10.36 (1H, N4H), 9.02–7.59 (4H, pyH), 7.62–7.02 (5H, NPh), 2.64 (3H, CCH_3_). ^13^C (DMSO, ppm): 177.2 (C=S), 144.5 (C=N), 142.3–123.2 (py), 140.1–119.9 (NPh), 20.1 (CCH_3_).

*[Pt(4PLO)Cl]* (**6**): Orange (0.44 g, 77%). FAB^+^ MS [*m*/*z*, assignment]: 480 [Pt(4PLO)]^+^, 286 [4PLO]^+^. C_14_H_13_ClN_4_OPtS (515.88): calcd. C 32.6, H 2.54, N 10.9, S 6.2; found C 32.6, H 2.7, N 10.8, S 6.2. Selected IR data (ν_max_/cm^−1^): 3315 w ν(NH), 1601 m, 1560 m, 1496 s, 1485 s, 1440 s [ν(C=N) + ν(C=C)], 1325 m [ν(C=S) + ν(C=N)], 1257 w, 1238 w, 1186 m ν(NO), 1141 w, 1099 w, 1072 w, 1041 m ν(NN), 848 w ν(C=S), 754 m. Far-IR (Nujol, ν/cm^−1^): 440 w ν(Pt–N), 427 w ν(Pt-O), 325 s ν(Pt–S), 320 sh ν(Pt–Cl). UV-Vis, (λ_max_, cm^–1^): 25,940, 24,271, 22,980, 17,241, 15,661. Conductance (Λ_m_/μS cm^−1^) in DMF: 9. ^1^H NMR (DMSO, ppm): 10.36 (1H, N4H), 9.02–7.34 (4H, pyH), 7.37–7.6.91 (5H, NPh), 2.65 (3H, CCH_3_). ^13^C (DMSO, ppm): 174.2 (C=S), 144.5 (C=N), 142.3–123.2 (py), 140.1–119.9 (NPh), 20.1 (CCH_3_). ^195^Pt NMR (DMSO, δ ppm): −2420.4.

*Pd(4DMLO)Cl]* (**7**): Yellow (0.26 g, 72%). FAB^+^ MS [*m*/*z*, assignment]: 378 [Pd(4DMLO)Cl]^+^, 345 [Pd(4DMLO)]^+^. C_10_H_13_ClN_4_OPdS (379.17): calcd. C 31.7, H 3.46, N 14.8, S 8.5; found C 31.8, H 3.4, N 14.6, S 8.5. Selected IR data (ν_max_/cm^−1^): 1562 w, 1541 m, 1494 m, 1408 s [ν(C=N) + ν(C=C)], 1369 m [ν(C=S) + ν(C=N)], 1292 m, 1259 m, 1228 w, 1188 m ν(NO), 1168 w, 1138 m, 1101 w, 1076 w, 1045 m ν(NN), 999 w, 916 m, 827 w ν(C=S), 771 m. Far-IR (Nujol, ν/cm^−1^): 340 sh ν(Pd–S), 318 s ν(Pd-Cl). UV-Vis, (λ_max_, cm^–1^): 26,420, 24,271, 22,805. Conductance (Λ_m_/μS cm^−1^) in DMF: 11. ^1^H NMR (DMSO, ppm): 8.82–7,68 (4H, pyH), 3.37, 3.18 (6H, NCH_3_), 2.26 (3H, CCH_3_). ^13^C (DMSO, ppm): 183.7 (C=S), 138.5 (C=N), 125.0–123.0 (py), 34.6 (NCH_3_), 14.4 (CCH_3_).

*[Pt(4DMLO)Cl]* (**8**): Orange (0.33 g, 71%). FAB^+^ MS [*m*/*z*, assignment]: 468 [Pt(4DMLO)Cl]^+^, 431 [Pt(4DMLO)]^+^. C_10_H_13_ClN_4_OPtS (467.83): calcd. C 25.7, H 2.80, N 12.0, S 6.9; found C 25.8, H 3.0, N 11.7, S 7.0. Selected IR data (ν_max_/cm^−1^): 1560 w, 1537 m, 1493 m, 1481 m, 1415 s [ν(C=N) + ν(C=C)], 1300 m, 1261 m [ν(C=S) + ν(C=N)], 1224 w, 1188 m, 1168 w, 1140 w ν(NO), 1076 w, 1045 m ν(NN), 1004 w, 918 m, 829 w ν(C=S), 769 m. Far-IR (Nujol, ν/cm^−1^): 441 w ν(Pt–N), 336s ν(Pt–S), 318 m ν(Pt–Cl). UV-Vis, (λ_max_, cm^–1^): 26,580, 22,624, 19,920, 16,934. Conductance (Λ_m_/μS cm^−1^) in DMF: 20. ^1^H NMR (DMSO, ppm): 8.99–7,61 (4H, pyH), 3.39, 3.34 (6H, NCH_3_), 2.26 (3H, CCH_3_). ^13^C (DMSO, ppm): 183.6 (C=S), 139.0 (C=N), 126.1–123.7 (py), 34.4 (NCH_3_), 14.2 (CCH_3_). ^195^Pt NMR (DMSO, δ ppm): −2420.4.

### 3.4. Single-Crystal X-ray Diffraction

Crystals were mounted on glass fibers, and these samples were used for data collection. The data for **3** and **5** were collected using an Enraf Nonius CAD4 automatic diffractometer [28] and corrected for absorption using psi-scan corrections [29]. The structures were solved via the direct methods [30], which revealed the position of all non-hydrogen atoms. These atoms were refined on *F*^2^ via a full-matrix least squares procedure using the anisotropic displacement parameters [30]. The hydrogen atoms were located from difference syntheses or in their calculated positions (C–H, 0.93–0.97 Å), and were refined using a riding model. Atom scattering factors were taken from the International Tables for Crystallography [31]. Molecular graphics were generated using DIAMOND software (Version 5.0.2) [32]. The crystal data, experimental procedures and refinement outcomes are summarized in Table 1.

### 3.5. General Procedure for the Suzuki–Miyaura Coupling Reactions

All the reactions were carried out according to the procedure described by Bhattacharya et al. [33]. A round-bottomed flask equipped with a reflux condenser was charged in air with 1 mole percent of complex **3** or **5**, Na_2_CO_3_ (0.212 g, 2 mmol), phenylboronic acid (0.183 g, 1.5 mmol) and aryl bromide (1.0 mmol) with DMF (3 mL). The flask was placed in a preheated oil bath at 120 °C for 24 h and then allowed to cool to room temperature. After the addition of water (20 mL) and repeated extraction using dichloromethane, the organic phase was washed with water, dried over Na_2_SO_4_ and filtered. The solvent was removed under a vacuum. The residue was dissolved in CDCl_3_ and analyzed via ^1^H NMR spectroscopy. The percentage conversion was determined from the remaining aryl halide.

## 4. Conclusions

Four complexes of Pd(II) and four of Pt(II) were prepared via a reaction between 2-acetylpyridine N-oxide-, 4N-methyl-, -ethyl-, -phenyl- and -dimethyl-substituted TSCs and K_2_[PdCl_4_] or K_2_[PtCl_4_]. The new complexes were characterized by applying a range of spectroscopic techniques. In the case of the Pd(II) complexes with 4N-ethyl- and 4N-phenyl-substituted TSCs, their molecular structure was determined via X-ray diffraction of single crystals. Both the spectroscopic studies and structural analysis demonstrate that TSCs behave like monoanionic pincer-type ONS ligands, coordinating metals through the oxygen atoms of pyridyl N-oxide, azomethine nitrogen and thiolate sulfur, thereby originating a square planar coordination geometry. This results in the formation of two chelate rings of 5 and 6 members, thereby completing the coordination sphere with the occupation of a chloride ion in a trans position relative to the nitrogen atom. The potential for palladium complexes to be candidates in different organic transformations has been evaluated in light of their ability to form pincer-like complexes. The complexes, whose structures were determined via X-ray diffraction, have been identified as the precursors of homogeneous catalysts for the Suzuki–Miyaura cross-coupling reaction of phenylboronic acids and aryl halides. This reaction has been shown to result in an appreciable yield. In the case of Pt(II) complexes with 4N-phenyl- and 4N-dimethyl-substituted TSCs, the sole signal observed in the ^195^Pt NMR spectrum indicates an appreciable stability against the solvolysis processes.

## Data Availability

The data presented in this study are available in the article and Appendix A.

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
