# Peer review of "Palladium(II) and Platinum(II) Complexes Bearing ONS-Type Pincer Ligands: Synthesis, Characterization and Catalytic Investigations"

_molecules, 2024, doi:10.3390/molecules29143425_

Round 1

Reviewer 1 Report

Comments and Suggestions for Authors

This is a nice work.  1.     The catalytic degradation efficiency should be compared with these related catalysts. 2. Give sound evidence for mechanism.

Author Response

Referee: 1

Comments

This is a nice work.

Reply: Thank you very much for your opinion and suggestions to improve the manuscript.

  1. The catalytic degradation efficiency should be compared with these related catalysts.

Reply: Thank you for bringing this to our attention, but by definition, TOF values are useful for evaluating a catalyst under certain specific conditions, but not for comparing different catalysts.

  1. Give sound evidence for mechanism.

Reply: According to their suggestion, a new paragraph on the mechanism has been included containing the reference to possible mechanism based on our results, and literature reports.

Reviewer 2 Report

Comments and Suggestions for Authors

The authors reported the synthesis of eight new Pd(II) and Pt(II) complexes by subjecting 2-acetylpyridine N-oxide derivatives to reactions with K2[PdCl4] or K2[PtCl4]. These products have been fully characterized including X-ray diffraction analysis. Two of these Complexes have been examined as catalysts in Suzuky-Miyaura cross-coupling reactions of aryl bromides with phenylboronic acid, although turnover numbers are not so high under harsh conditions. Publication of this work in Molecules is recommended. One major question needs to be discussed: boronic acid was utilized in this work. However, it’s wondered if boronic ester- bis (pinacolato)diboron works better?

please address how to improve catalytic turnovers and add at least one more substrate in Suzuki coupling, e.g. R = Ph in Table 3.

Author Response

REVIEWER REPORT(S):

Referee: 2

Comments

The authors reported the synthesis of eight new Pd(II) and Pt(II) complexes by subjecting 2-acetylpyridine N-oxide derivatives to reactions with K2[PdCl4] or K2[PtCl4]. These products have been fully characterized including X-ray diffraction analysis. Two of these Complexes have been examined as catalysts in Suzuky-Miyaura cross-coupling reactions of aryl bromides with phenylboronic acid, although turnover numbers are not so high under harsh conditions. Publication of this work in Molecules is recommended. One major question needs to be discussed: boronic acid was utilized in this work. However, it’s wondered if boronic ester- bis (pinacolato)diboron works better?

Reply: Thank you for bringing this to our attention. One of the objectives of the present investigation was to evaluate the catalytic capacity of TSCs, not to study the behavior of different boron compounds in the Suzuki-Miyaura reaction. However, new studies on the catalytic activity of palladium(II) complexes with thiosemicarbazones using other boron compounds will be carried out in the future. In any case, the conclusion from a recent analysis of the seven major classes of boron reagents is that there will probably never be a single boron species that is the reagent of choice for all SM couplings (DOI: 10.1039/c3cs60197h).

Reviewer 3 Report

Comments and Suggestions for Authors

The Castiñeiras and co-workers have presented the synthesis and full characterization of new complexes with ONS-type pincer ligands of Palladium and Platinum. Additionally, they tested the new palladium complexes in a series of Suzuki reactions, disclosing their high activity. This reviewer has uploaded a PDF file with some notes to the text. This review supports the publication of the present paper after the minor revisions suggested in the PDF and the one here suggested.

Figure 1. Please change the preset Xray with a more appropriate ORCID view.

Author Response

REVIEWER REPORT(S):

Referee: 3

Comments

The Castiñeiras and co-workers have presented the synthesis and full characterization of new complexes with ONS-type pincer ligands of Palladium and Platinum. Additionally, they tested the new palladium complexes in a series of Suzuki reactions, disclosing their high activity. This reviewer has uploaded a PDF file with some notes to the text (molecules-3114097-review). This review supports the publication of the present paper after the minor revisions suggested in the PDF and the one here suggested.

Reply: Thank you very much for your opinion and suggestions. Typing mistakes and other minor flaws have been corrected.

Figure 1. Please change the preset X ray with a more appropriate ORTEP view.

Reply: Corrected, thanks. Figure 1 now includes an ORTEP drawing, and the figure caption has been corrected accordingly.